# Clinical associations with a polygenic predisposition to benign lower white blood cell counts

Jonathan D. Mosley [1,2] ✉, John P. Shelley[2], Alyson L. Dickson [1], Jacy Zanussi[1], Laura L. Daniel [1], Neil S. Zheng[1,3], Lisa Bastarache[2], Wei-Qi Wei [2], Mingjian Shi [2], Gail P. Jarvik[4,5], Elisabeth A. Rosenthal [5], Atlas Khan [6], Alborz Sherafati[7], Iftikhar J. Kullo [7], Theresa L. Walunas [8], Joseph Glessner [9], Hakon Hakonarson [10], Nancy J. Cox[1], Dan M. Roden [1,2,11], Stephan G. Frangakis[12], Brett Vanderwerff[13], C. Michael Stein[1,11], Sara L. Van Driest[1,14], Scott C. Borinstein [14], Xiao-Ou Shu[1,15], Matthew Zawistowski [13], Cecilia P. Chung[16] & Vivian K. Kawai[1]

Polygenic variation unrelated to disease contributes to interindividual variation in baseline white blood cell (WBC) counts, but its clinical significance is uncharacterized. We investigated the clinical consequences of a genetic predisposition toward lower WBC counts among 89,559 biobank participants from tertiary care centers using a polygenic score for WBC count (PGS_WBC) comprising single nucleotide polymorphisms not associated with disease. A predisposition to lower WBC counts was associated with a decreased risk of identifying pathology on a bone marrow biopsy performed for a low WBC count (odds-ratio = 0.55 per standard deviation increase in PGS_WBC [95%CI, 0.30−0.94], p = 0.04), an increased risk of leukopenia (a low WBC count) when treated with a chemotherapeutic (n = 1724, hazard ratio [HR] = 0.78 [0.69−0.88], p = 4.0 × 10⁻⁵) or immunosuppressant (n = 354, HR = 0.61 [0.38−0.99], p = 0.04). A predisposition to benign lower WBC counts was associated with an increased risk of discontinuing azathioprine treatment (n = 1,466, HR = 0.62 [0.44−0.87], p = 0.006). Collectively, these findings suggest that there are genetically predisposed individuals who are susceptible to escalations or alterations in clinical care that may be harmful or of little benefit.

White blood cell (WBC) counts (the number of WBCs present within a volume of blood) are routinely measured in clinical settings to survey health, ascertain for drug toxicities and identify causes of illness. The counts are evaluated with respect to a reference interval of values expected in a healthy population[1,2], and a measurement that falls outside of the interval may prompt investigations to exclude conditions such as infections, diseases of the bone marrow, autoimmune disease, and toxicities due to medications such as chemotherapeutics and immunosuppressants[3]. A low WBC count may also prompt clinical action due to concerns that an individual may have an immunodeficiency that could limit an effective response to infections[4].

A genetic predisposition toward benign lower WBC counts can impact clinical care. For instance, the rs2814778-CC genotype is common among individuals of African ancestry and is associated with lower WBC counts in the absence of underlying disease[5,6]. Carriers of this genotype are more likely to undergo diagnostic investigations, including a bone marrow biopsy, and to have medications stopped due

to concerns for toxicity[7–10]. These actions are driven, in part, by the use of WBC count reference ranges that are not calibrated to this genotype[11,12]. While the rs2814778-CC genotype is not prevalent among European ancestry (EA) populations, numerous common single nucleotide polymorphisms (SNPs) associated with WBC count variation have been identified in this group[13,14]. Whether a polygenic predisposition toward benign lower WBC counts could have similar clinical consequences in this population is unknown.

We constructed a polygenic score for WBC counts (PGS_WBC) which measures the burden of SNPs associated with WBC count that an individual carries. SNPs located near loci associated with clinically significant diseases within the differential diagnosis of a low WBC count were excluded to ensure that the PGS_WBC measures benign WBC count variation. We examine a diverse range of clinical outcomes and settings to characterize the consequences of a polygenic predisposition to benign lower WBC counts.

## Results

### Development and validation of a benign PGS_WBC
We developed a polygenic risk score for WBC count (PGS_WBC) using SNP weightings derived from a large genome-wide association study (GWAS) of WBC counts[15]. A linkage-disequilibrium reduced ($r^2 < 0.01$) set of independent SNPs was selected ($p < 5 \times 10^{-6}$, minor allele frequency > 0.01, imputation $r^2 \geq 0.7$). To ensure that SNPs associated with clinically significant diseases were not included in the PGS_WBC, SNPs located in the major histocompatibility complex (MHC) genomic region, which is associated with multiple autoimmune diseases[16], or near loci associated with hematological malignancies or systemic lupus erythematosus[17] were excluded (see "Methods" section for full details). After exclusions, there were 1739 SNPs in the PGS_WBC. The PGS_WBC was normalized to have a mean of 0 and a standard deviation of 1, and a lower PGS_WBC value reflects a polygenic predisposition to lower WBC counts. Thus, an inverse association between the PGS_WBC and an outcome indicates that a predisposition to lower counts increases the risk of the outcome.

To verify that the PGS_WBC did not associate with clinically significant diseases that are in the differential diagnosis of a low WBC count, we tested for associations between the PGS_WBC and hematological malignancies ($n = 15$ diagnoses) and autoimmune diseases ($n = 21$ diagnoses) that were prevalent among 71,078 European Ancestry participants from BioVU, a DNA biobank linked to a de-identified electronic health record. There was one nominal inverse association ($p < 0.05$) with a diagnosis of psoriasis vulgaris, an autoimmune skin condition (Supplementary Fig. 1 and Supplementary Table 1).

There were also 4 nominal positive associations ($p < 0.05$) with hematological malignancies, suggesting a that predisposition to higher WBC counts could be associated with the risk of these diagnoses. To determine whether associations were driven by SNPs with a relatively larger effect size, SNPs in the PGS were separated into 5 quintiles based on the distribution of effect sizes, and a PGS for each quintile was constructed. Only the PGS for the 4th quintile (representing SNPs in the 60–80th percentile) had more positive associations than expected by chance ($n = 8$ phenotypes, $p = 0.001$, based on permutation analyses) (Supplementary Table 2). In addition, SNPs in this quintile had more nominal ($p < 0.05$) associations with 4 hematological phenotypes than would be expected by chance ($p < 0.05$ based on binomial expectations) (Supplementary Table 3). Collectively, these results suggest that SNPs of larger effective sizes did not drive the positive hematological associations, but might suggest that the PGS_WBC may comprise SNPs of modest effect size that have weak positive associations with hematological malignancies.

We examined associations between the PGS_WBC and a diverse range of clinical outcomes measured in different clinical settings (Fig. 1).

### Association between the PGS_WBC and low measured WBC and markers of clinical activity
To characterize the relationship between the PGS_WBC and measured WBC counts, we identified 11,694 BioVU participants (6931 females [59%]; mean age 57 [s.d. 17] years) without a hematological malignancy and who had 1 or more WBC count measurements collected in a primary care setting during a routine health maintenance exam. The PGS_WBC was positively correlated (partial correlation = 0.29, adjusted for age, sex, and 5 PCs) with measured median WBC counts (Fig. 2a). We also investigated the associations between the PGS_WBC and an individual's lowest measured WBC count and found the same association Supplementary Fig. 2. When WBC count measurements are reported in clinical settings, the results are presented in conjunction with the reference range for the assay used to measure the count. There were 623 participants who had at least one WBC count that fell below the lower reference range value (i.e., these participants had a value that would be designated as a clinical outlier). The PGS_WBC was inversely associated with the outcome of having a WBC count below the lower reference range value (odds-ratio [OR] = 0.57 [95% CI: 0.52–0.62] per s.d. increase in the PGS_WBC, $p < 2 \times 10^{-16}$) (Fig. 2b). The model demonstrated a good fit to the data (Hosmer–Lemeshow $p = 0.66$) Supplementary Fig. 3.

ICD codes are assigned to patients to indicate the diagnoses that were addressed during a clinical encounter[18]. If a patient has been assigned an ICD code for a low WBC count, it indicates that the clinical provider diagnosed the patient with a low WBC count. There were 379 (3.2%) participants assigned an ICD code for a low WBC count. The PGS_WBC was significantly associated with being assigned an ICD code for a low WBC count among BioVU participants (OR = 0.62 [0.56–0.69], $p < 2 \times 10^{-16}$). Thus, a polygenic predisposition to lower WBC counts increased the likelihood of receiving a clinical diagnosis of a low WBC count. We also tested for an association in an independent set of participants from the eMERGE network, a consortium of institutions with EHR-linked biobanks. Among eMERGE participants who did not have a history of a hematological malignancy ($n = 18,217$), there were 256 (1.4%) participants who had an ICD code for a low WBC count. The PGS_WBC was associated with this outcome (OR = 0.74 [0.67–0.82], $p = 2.0 \times 10^{-9}$).

In sensitivity analyses, we examined a PGS that excluded SNPs that had nominal associations with a hematological malignancy in the BioVU population. Associations with having an ICD code for a low WBC count were not changed with these exclusions (Supplementary Table 4).

In sum, a polygenic predisposition toward lower WBC counts was associated with lower measured WBC counts, an increased likelihood of having a count below the clinical reference range, and having an ICD code for a low count.

### Association between the PGS_WBC and bone marrow biopsies for low WBC
A bone marrow biopsy is an invasive procedure to determine whether a hematological abnormality, such as a low WBC count, is due to an underlying disease in the bone marrow. Biopsy reports include a clinical indication, which lists the clinical concerns of the hematologist that prompted the biopsy. It has been previously observed that a benign WBC-lowering genotype was associated with both the clinical indication and outcomes of a bone marrow biopsy among individuals of African ancestry[9]. We examined whether the PGS_WBC behaved similarly in European ancestry populations.

There were 922 BioVU participants without a prior history of a hematological malignancy who underwent a first bone marrow biopsy. We tested whether the PGS_WBC was associated with undergoing a biopsy due to a clinical concern for a low WBC count. There were 117 participants biopsied for this reason, and the PGS_WBC was associated with this outcome (OR = 0.56 [0.45–0.68], $p = 1.8 \times 10^{-8}$). This result indicates that a polygenic predisposition to a lower WBC count increased the likelihood that a biopsy was performed due to concerns for a low count.

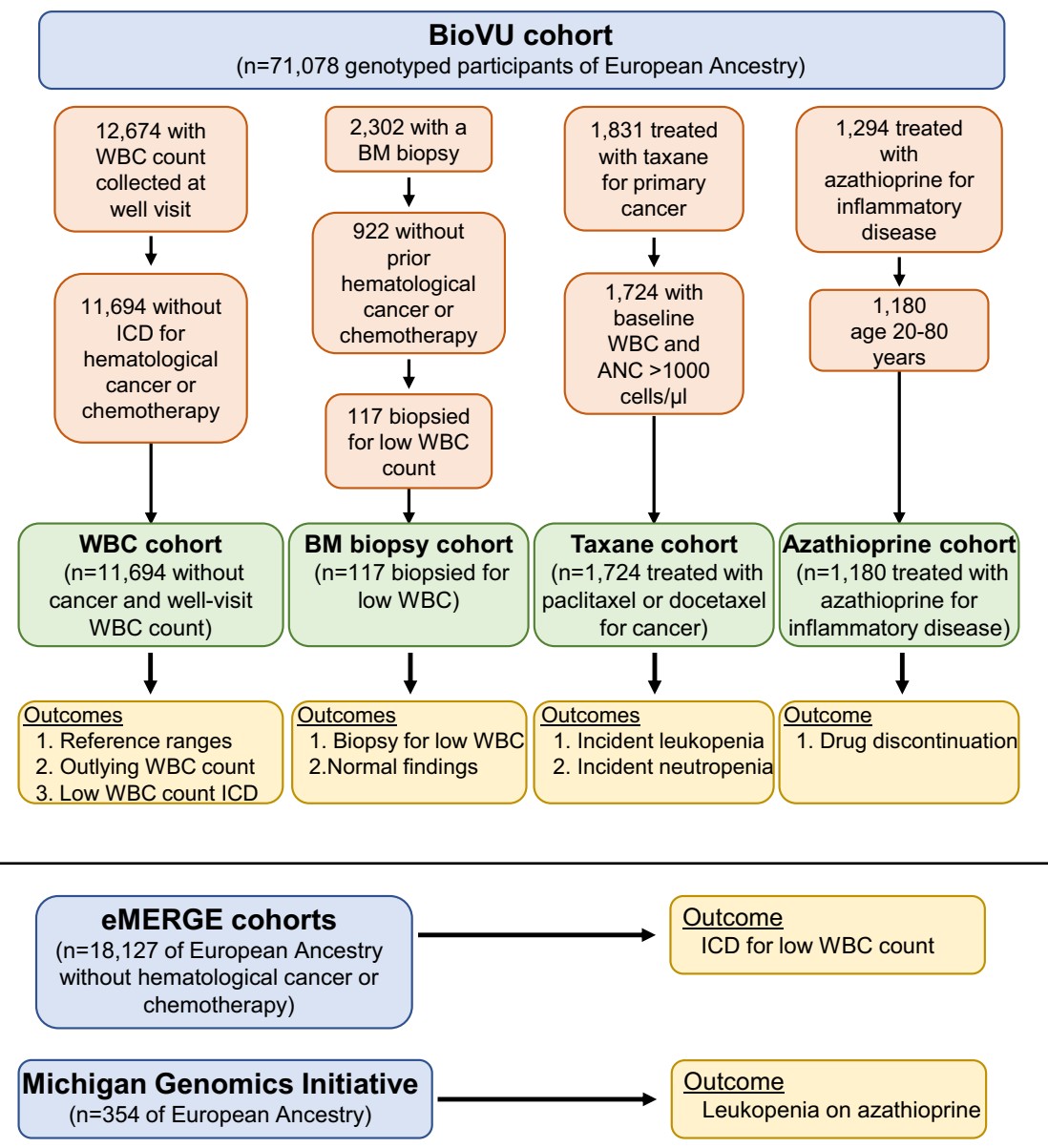

**Fig. 1 | Overview of the study populations and analyses.** WBC white blood cell, BM bone marrow, ANC absolute neutrophil count, ICD International Classification of Disease code.

Among the 117 participants biopsied for a low WBC count, 64 (55%) were female and the mean age was 48 (s.d. 25) years (Supplementary Table 5). Bone marrow pathology was identified in 35 (30%) biopsies. Biopsies which showed pathology were more likely to have a second hematological comorbidity (e.g. anemia or a low platelet count) noted in the indication, as compared to those without pathology ($n = 29$ [83%] vs $n = 37$ [45%]) (Supplementary Table 5). Those without pathology in their bone marrow had $PGS_{WBC}$ values that were more skewed toward the lower ranges compared to those with pathology (Fig. 3a). There was not a significant difference in the mean $PGS_{WBC}$ value between those participants with an abnormal biopsy and the unbiopsied BioVU population (difference = −0.11 standard deviations [95% CI: −0.43−0.22], $p = 0.53$). The $PGS_{WBC}$ was associated with a finding of no pathology, after adjusting for other hematological comorbidities (OR = 0.55 [0.30−0.94], $p = 0.04$) (Fig. 3b). Associations with biopsy outcomes were not changed when excluding SNPs from the PGS that had nominal associations with a hematological malignancy in BioVU (Supplementary Table 6).

We determined the proportion of participants who underwent a biopsy and whose WBC count at the time of biopsy would be considered within the normal range of a genotype-informed reference range. For these analyses, the normal range was defined as a value above the 2.5th percentile of the distribution of WBC counts observed among individuals with a similar $PGS_{WBC}$ value (see "Methods" section). All individuals whose WBC count at the time of biopsy was between 3500 and 4000 cell/μL fell within their genetically expected ranges, regardless of biopsy outcome (Table 1, Fig. 3c, d). However, among individuals whose WBC count was <=3500 cells/μL, only individuals with a normal biopsy result had counts that fell within genetic ranges. For instance, among individuals with WBC count between 3000 and 3500 cells/μL, 9 of 13 (69%) individuals with a normal biopsy, versus 0 of 6 (0%) with an abnormal biopsy, fell within expected ranges. These results are consistent with the lower $PGS_{WBC}$ values observed in the normal biopsy group (Fig. 3a). No individuals with a WBC count <2500 cells/μL fell within their genetic ranges.

In sum, a polygenic predisposition toward lower WBC counts was associated with an increased likelihood of having a bone marrow

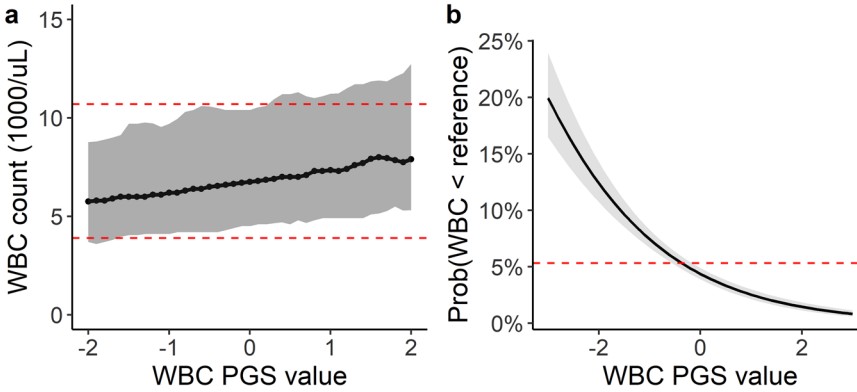

**Fig. 2 | PGS associations with measured WBC counts among 11,694 BioVU participants. a** Ranges of observed median WBC counts by PGS_WBC value. Ranges summarize WBC counts within sequential windows (±0.2 s.d.) across the range of the PGS_WBC. The dark line is the median value, and the gray ribbon represents the 5th–95th percentiles of the range. The dashed red lines denote the upper and lower clinical reference ranges for the clinical assay used to measure the WBC count. **b** Predicted probability (with standard errors) of having a WBC count that falls below the lower clinical reference value across a range of PGS_WBC values ($n = 623$ events). Probabilities are based on a logistic regression model adjusted for age and sex. The dashed red line is the average probability for the entire population.

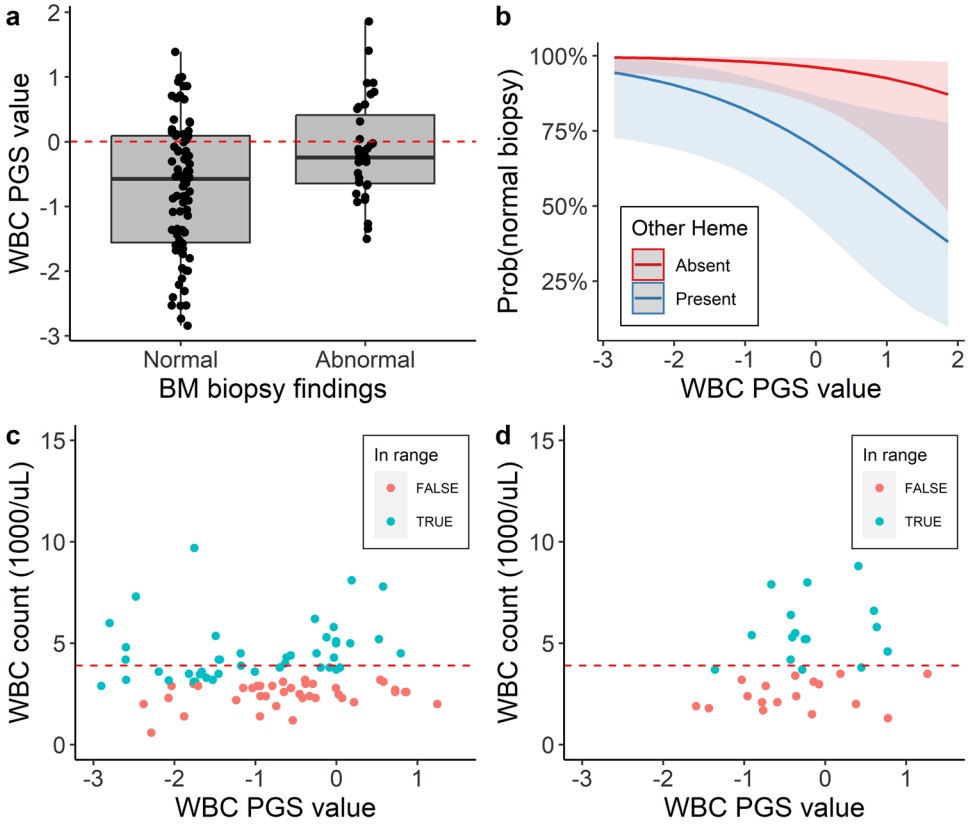

**Fig. 3 | PGS associations with bone marrow biopsy outcomes. a** Distribution of the PGS_WBC values among 117 BioVU participants who underwent a bone marrow biopsy for a clinical indication that included a low WBC count. Results are stratified by whether the biopsy identified pathology (abnormal, $n = 35$) or not (normal, $n = 82$). Box plots show the median value, interquartile range (gray region), and maximum and minimum values (whiskers) for each stratum. The dashed red line indicates the median PGS_WBC value for the overall BioVU population. **b** Predicted probabilities (with standard errors) of a normal biopsy finding. Results are stratified by whether the indication for the biopsy included other hematological abnormalities (labeled as present or absent) in addition to a low WBC count. Probabilities are based on a multivariable logistic regression model, adjusted for age, sex, and presence of a hematological abnormality. **c, d** Scatterplots showing the WBC count at the time of biopsy versus the PGS_WBC values for participants with a **c** normal or **d** abnormal biopsy result. Points are colored to denote whether the observed WBC count fell within the range (i.e. above the 2.5th percentile of the distribution) of WBC counts observed among individuals in the WBC cohort whose PGS_WBC value was within 0.2 standard deviations. The dashed red line denotes the lower bound of the clinical reference ranges.

biopsy that was performed to investigate a low WBC count and a reduced likelihood of identifying pathology on the biopsy. The PGS_WBC may also preferentially reclassify individuals with modestly low WBC counts and without underlying bone marrow disease whose counts.

**Association between the PGS_WBC and drug-induced leukopenia**
Among individuals of African ancestry, benign neutropenia associates both with an increased risk of a low WBC count due to a medication (i.e. drug-induced leukopenia) and medication discontinuation due to a concern for low WBC counts[10,19]. We examined whether the PGS_WBC

**Table 1 | Proportions of individuals whose WBC count at biopsy fell within a genotype-informed range**

| WBC range (×1000 cells/μL)[a] | Biopsy outcome: Normal | | | Biopsy outcome: Abnormal | | |
|---|---|---|---|---|---|---|
| | Number in range | Number within genotype-informed range[b] | Percentage within range | Number in range | Number within genotype-informed range[b] | Percentage within range |
| >3.5–4.0 | 10 | 10 | 100 | 3 | 3 | 100 |
| >3.0–3.5 | 13 | 9 | 69.2 | 6 | 0 | 0 |
| >2.5–3.0 | 18 | 1 | 5.6 | 2 | 0 | 0 |
| 0–2.5 | 18 | 0 | 0 | 10 | 0 | 0 |

[a]WBC count at the time of bone marrow biopsy.
[b]Individuals whose WBC count fell above the 2.5th percentile of the distribution of WBC counts observed among individuals in the WBC cohort whose $PGS_{WBC}$ value was within 0.2 standard deviations. WBC count distributions were constructed based on the lowest observed WBC count among an individual.

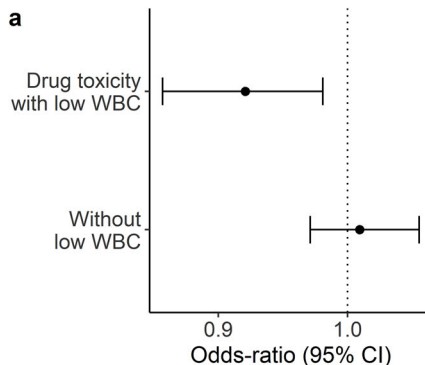

**Fig. 4 | PGS associations with pharmacogenomic outcomes. a** Odds-ratio and 95% confidence interval of having an ICD code for toxicity from an antineoplastic or immunosuppressive medication with ($n = 985$) or without ($n = 2273$) a clinical concern for a low WBC count among 71,078 BioVU participants. Odds-ratios are from a logistic regression model, adjusted for age, sex, and principal components. **b** Kaplan–Meier plot for a WBC count < 3000 cells/μL (leukopenia) ($n = 266$ events) after initiating treatment with taxanes among 1724 BioVU participants with cancer. The $PGS_{WBC}$ strata are Low (<1 s.d. below the mean), Middle (≥−1 s.d. and ≤1 s.d.), High (>1 s.d.).

was similarly associated with drug side effects for two classes of drugs that can commonly cause drops in WBC counts: chemotherapeutics (antineoplastics) and immunosuppressants.

We first examined whether there was an association with ICD-based billing codes related to adverse effects from these classes of medications. Since these medications can be associated with a broad range of toxicities, we specifically examined whether the $PGS_{WBC}$ was associated with having an ICD code for toxicity due to chemotherapeutic or immunosuppressant medications entered on the same day that the provider wrote a clinical note that mentioned a low WBC count. In the BioVU population ($n = 71,078$), there were 985 (1.4%) participants who met this case definition and the $PGS_{WBC}$ was associated with this outcome (OR = 0.92 [0.86–0.98], $p = 0.01$) (Fig. 4a). However, there was no association with those ICD codes in the absence of a clinical mention of a low WBC count ($n = 2273$ cases, OR = 1.01 [0.97–1.06], $p = 0.60$).

To probe this association further, we identified 1724 BioVU participants (917 [53%] female, 60 [12] years) who received treatment for cancer with the taxane class of chemotherapeutic medications[20] (Supplementary Table 7). The mean baseline (pre-treatment) WBC count was 8200 (s.d. 3900) cells/μL. The $PGS_{WBC}$ was associated with baseline (pre-treatment) count (change in log-transformed WBC count = 0.062 [0.044–0.080], $p = 3.9 \times 10^{-11}$). In the first cycle of treatment, 266 (16%) participants developed an incident drug-induced leukopenia, defined here as a WBC count < 3000 cells/μL. The $PGS_{WBC}$ was significantly associated with time to leukopenia after adjusting for treatment dose and duration (hazard ratio [HR] = 0.78 [0.69–0.88], $p = 4.0 \times 10^{-5}$) (Fig. 4b). There was a similar association for the outcome of time to drug-induced neutropenia (absolute neutrophil count < 1500 cells/μL) (HR = 0.80 [0.69–0.91], $p = 0.0006$) (Supplementary Fig. 4).

Similar results were seen in an independent cohort of 354 participants (203 [57%] female, 44 [17] years) from the Michigan Genomics Initiative who were treated with the immunosuppressant azathioprine for an autoimmune disease (Supplementary Table 8). The WBC count at azathioprine initiation was 8800 [3700] cells/μL. The $PGS_{WBC}$ was significantly associated with time to an incident WBC count < 3000 cells/μL with treatment (HR = 0.61 [0.38–0.99], $p = 0.04$) (Supplementary Fig. 5).

## Association between the $PGS_{WBC}$ and medication discontinuation

To determine whether the $PGS_{WBC}$ was associated with cessation of medications due to a provider's concern for a low WBC count, we examined a cohort of 1180 BioVU participants (787 [67%] female, 47 [15] years) treated with azathioprine for an autoimmune disease (Supplementary Table 9)[10]. The $PGS_{WBC}$ was inversely associated with WBC count at azathioprine initiation (change in log-transformed WBC count per s.d. increase in the $PGS_{WBC}$ = 0.050 [0.026–0.073], $p = 3.7 \times 10^{-5}$), but not the change in WBC count during the course of treatment (change in log-transformed WBC count = 0.041 [0.270–0.350], $p = 0.79$). Azathioprine was discontinued in 34 (3%) participants due to a clinical concern for a low WBC count. The $PGS_{WBC}$ was associated with time to medication discontinuation (HR = 0.62 [0.44–0.87], $p = 0.006$, adjusted for baseline dose) (Fig. 5). Thus, a polygenic predisposition toward lower WBC counts was associated with an increased likelihood of stopping an immunosuppressive medication due to concern for a lower WBC count.

In sum, a polygenic predisposition toward lower WBC counts was associated with an increased likelihood of having ICD codes for drug toxicity related to a low WBC count, developing leukopenia with treatment, and medication discontinuation.

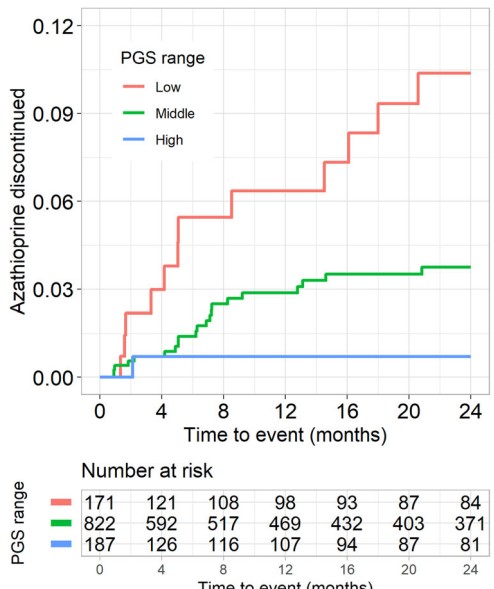

**Fig. 5 | Azathioprine discontinuation.** Kaplan–Meier plot for discontinuation of azathioprine due to clinical concern of a low WBC count related to medication toxicity (n = 34 events) among 1180 BioVU participants with autoimmune disease. The PGS$_{WBC}$ strata are Low (<1 s.d. below the mean), Middle (≥−1 s.d. and ≤1 s.d.), High (>1 s.d.).

## Discussion

We examined whether a polygenic predisposition to a lower WBC count was associated with clinically significant endpoints. This predisposition was associated with an increased likelihood of having a WBC count falling outside the reference range and receiving an ICD code specifically noting an outlying value. It was also associated with an increased likelihood that a hematologist would list a low WBC count as a reason for performing a bone marrow biopsy, but a decreased likelihood that that such biopsies identified disease within the bone marrow. Genotype-informed reference ranges also demonstrated that individuals without underlying bone marrow disease were more likely to fall within expected ranges, as compared to those with pathology. Finally, a predisposition to lower a WBC count increased the likelihood of drug-induced leukopenia with treatment using chemotherapeutic or immunosuppressant therapies and increased the likelihood of cessation of azathioprine due to clinical concern for a low WBC count. Collectively, these analyses demonstrate that genetic variation which does not contribute to disease risk, is associated with clinical consequences such as undergoing diagnostic procedures or altering therapeutic regimens.

A broad range of heritable biomarkers, including WBC counts, are measured in clinical settings to guide clinical care. Many of these biomarkers are not direct mediators of disease but fluctuate in response to acute and chronic illness. Thus, they are often used to characterize an illness, which helps establish a differential diagnosis. These biomarkers are paired with clinical reference ranges which are used to standardize the delivery of health care. These reference ranges are typically constructed such that 5% of healthy individuals will have a measure that lies outside of the interval and, thus, are outliers[1,21].

The standard clinical approach toward personalizing the interpretation of biomarkers is to serially measure them in individuals when they are healthy in order to define the range of values that they fluctuate in. The limitations of this approach are the costs of repeated measurements, especially if individuals are frequently changing providers. Furthermore, 5% of healthy individuals will have values outside standard population reference ranges, which can lead to uncertainty as to whether they may harbor occult underlying disease. This

uncertainty may prompt diagnostic evaluations, and is stressful to patients, as they may be subject to serial lab monitoring with no clearly specified endpoint.

The ability of genetic variation associated with benign biomarker variation to systematically disadvantage an identifiable subset of individuals is exemplified by the rs2814778-CC genotype, which is predominantly carried by individuals of African ancestry and underlies the clinical phenomenon of benign neutropenia[22]. The current study demonstrates that common polygenic variation associated with a diagnostic biomarker can push individuals toward the same clinical endpoints as a genotype with a large effect size.

Like WBC counts, many of the biomarkers that are used to guide clinical decision-making have a significant heritable component[23]. While polygenic variation impacts biomarker levels across the distribution, it is important to demonstrate that it can contribute to values that cross clinical decision thresholds, which typically lie near the tails of the distributions. We found that the PGS$_{WBC}$ was not only associated with an increased likelihood of having WBC counts below the lower limits of the standard lab reference ranges but was also associated with diagnostic ICD codes, indicating that the counts came to the attention of clinical providers and were prompting clinical actions.

Similar to the benign neutropenia genotype, a tool like the PGS$_{WBC}$ may prevent futile diagnostic odysseys or unnecessary alterations in clinical care. However, as compared to a discrete genotype, translating a measure of benign polygenic risk to the clinical setting is more nuanced, as the exposure is continuous. Furthermore, the approaches to clinical translation will vary by clinical scenario.

The bone marrow biopsy example highlights a scenario where WBC counts are used as a biomarker of occult hematological disease. In this scenario, it is important to know the expected ranges of WBC counts that a non-diseased individual fluctuates within. Genotype-informed reference ranges (such as those presented in Fig. 2a) could be informative for this purpose. As a simple proof-of-concept, we showed that applying reference intervals based on the PGS$_{WBC}$ identified more individuals without underlying bone marrow disease who fell within the expected WBC count range, as compared to individuals with underlying disease. Though only a modest proportion of individuals fell within expected ranges, it is important to note that each reclassification avoids an expensive and painful diagnostic procedure. An alternative approach could be to generate prior probabilities based on PGS$_{WBC}$ values that a given diagnostic workup (such as a bone marrow biopsy) will identify pathology when performed for an indication of a low WBC count (such as presented in Fig. 3b).

Another scenario is illustrated by the pharmacogenomic examples. Collectively, these studies indicate that the associations between PGS$_{WBC}$ and various outcomes are not driven by larger drug-induced drops in WBC counts, but rather a predisposition to WBC counts that fluctuate within ranges that cross clinical decision thresholds. The central clinical question in this scenario that remains to be addressed is whether the PGS$_{WBC}$ can identify individuals in whom it is safe to apply lower WBC count thresholds without increasing the risk of adverse outcomes, especially infection secondary to immunosuppression. If so, the PGS$_{WBC}$ may be informative to guide setting the lower limits of treatment thresholds in order to avoid systematic undertreatment of predisposed individuals. There is precedence for such a pharmacogenomic approach among individuals with benign neutropenia treated with clozapine, an antipsychotic that can lower WBC counts. Clozapine treatment guidelines permit lower WBC count discontinuation thresholds for individuals with this genotype[24].

Finally, these studies identify an alternative application of polygenic variation to clinical medicine. There has been considerable interest in using polygenic risk scores (PRS) in clinical settings to identify healthy individuals at risk for future disease[25,26]. This application of genetics leads to an escalation of clinical care for individuals

identified as high-risk, which always entails some degree of risk. A challenge of PRS-based clinical predictors is that they typically have modest discriminative capabilities with respect to identifying individuals who will develop incident disease[27,28]. With a modestly discriminative biomarker, the risks will often outweigh the benefits when clinical care is escalated. In contrast to the PRS prediction paradigm, a benign $PGS_{WBC}$ could have a role in de-escalating care and circumventing an otherwise futile diagnostic odyssey. This application of polygenic predictors is apt to have a more favorable risk-benefit ratio, as the default pathway for individuals with outlying values is an escalation in clinical care. Finally, the $PGS_{WBC}$ may provide a biologically motivated explanation for individuals who undergo an extensive diagnostic workup that does not identify an explanation for their leukopenia. Determining that they have a polygenic predisposition to lower WBC counts may allow them to avoid the costs and the need to undergo ongoing active surveillance that is often associated with clinical uncertainty.

A challenge of constructing a benign PGS for a clinical biomarker is determining which SNPs that may predispose to disease should be excluded from the predictor. For the $PGS_{WBC}$, we excluded SNPs in genomic loci that were associated with diseases that are in the differential diagnosis of a low WBC count, such as hematological malignancies. However, we did not exclude SNPs associated with intermediate phenotypes of hematological malignancies, such as clonal hematopoiesis, as these phenotypes often will not progress to disease and are also typically not associated with WBC count abnormalities in the absence of transformation to malignant disease[29–32]. More generally, when constructing a benign PGS for possible clinical application, it will be essential to define the clinical settings where the PGS will be used and to exclude all SNP variation associated with important diagnoses relevant to the setting and the biomarker. Defining the appropriate association $p$-value thresholds for exclusion can present challenges. Excluding a highly inclusive set of SNPs weakly associated with a large number of diseases will remove numerous SNPs from the PGS and degrade its performance. In contrast, only excluding SNPs associated with diseases at genome-wide significance may be too restrictive and risk misclassifying diseased individuals as benign outliers. Importantly, for any set of exclusion criteria, validation of the performance and utility of the PGS in real-world settings will be critical to determine its utility, safety, and limitations in practice.

There are strengths to the current study. In particular, it demonstrated a consistent pattern of associations across a diverse range of clinical outcomes and settings. There are also limitations. It is possible that this $PGS_{WBC}$ includes SNPs associated with diseases that decrease WBC counts. Mitigating this concern is the fact that the SNP weightings used to construct the $PGS_{WBC}$ were derived from a very large GWAS study where most individuals would not have manifest disease; thus, the influences of individuals with overt disease on SNP weightings would be expected to be small. However, we observed modest positive associations with several hematological diagnoses. These may have been a consequence of the analytical approaches used in the GWAS study that our SNP weightings were derived from. While the investigators attempted to exclude individuals with diagnosed hematological disease, they included individuals with WBC counts up to 200,000 cells/μL. As the probability of underlying disease with WBC counts of even 50,000 cells/μL is very high, there were likely individuals with manifest disease in their population. While the data were transformed to mitigate the impact of these outliers, their inclusion could explain the weak positive associations that we observed. The optimal approach to mitigating this issue would be to use more stringent exclusion ranges in GWAS that would be repurposed for clinical use. Importantly, we found that excluding SNPs with even nominal associations with hematological malignancies from the $PGS_{WBC}$ did not alter our underlying findings. It is also important to note that while we

excluded genomic regions around SNPs associated with significant clinical diseases within the differential diagnosis of low WBC counts, we relied on data from the GWAS catalog which may be incomplete and does not include disease-associated SNPs with lower levels of statistical significance. These analyses were restricted to European ancestries, as it has been previously demonstrated that a benign common genetic variant drives these same associations in African ancestries[9,10].

In conclusion, a polygenic predisposition toward benign lower WBC counts was associated with a diagnosis of leukopenia, undergoing diagnostic procedures, and medication discontinuation. Collectively, these studies describe a genetic tool that may help identify populations who are susceptible to escalations or alterations in clinical care or may have a role in personalizing biomarker WBC count reference ranges for the purpose of de-escalating unnecessary diagnostic investigations or alterations to clinical care.

## Methods
### Study populations
**BioVU.** Study populations were derived from Vanderbilt University Medical Center's (VUMC) DNA biobank resource (BioVU). BioVU comprises 270,000 consented participants and is constructed from discarded blood samples collected from consented individuals and linked to de-identified electronic health records (EHR)[33]. Participants are not compensated. The de-identified EHR (the Synthetic Derivative) captures a large portion of data available through the medical center's electronic health record[33]. Analyses were restricted to 71,078 participants of White European Ancestry with existing SNP genotyping. All BioVU studies were evaluated by the VUMC Institutional Review Board (IRB) and determined to be non-human subjects research. The following cohorts were derived from this population:

(1) WBC cohort: This cohort of 11,694 BioVU participants was used to characterize the relationship between the white blood cell count PGS and measured white cell counts. To minimize the likelihood that WBC counts were collected during an acute illness, the population was restricted to individuals with one or more WBC counts measured contemporaneously with an International Classification of Disease[34] (ICD)-9/10 code for a routine health exam (ICD-9: V70.9, V20, V20.1, V70, V70.0, V20.2; ICD-10 Z00.8, Z00.129, Z00.00, Z00.01, Z00.121). These codes are typically used in a primary care setting to denote a well-visit encounter. Participants with an ICD code for hematological malignancy, chemotherapy, or radiation therapy (ICD codes listed in Supplementary Data 1) were excluded, as these diagnoses or treatments can markedly impact WBC counts.

(2) Bone marrow biopsy cohort: Among 2302 BioVU participants with a biopsy pathology report in their clinical record, 1380 participants had an ICD code for hematological malignancy, blood transfusion, organ transplant, or chemotherapy/radiation prior to their first biopsy and were excluded[12]. Additional exclusions included: the absence of a text phrase pertaining to a low WBC count in the clinical indication portion of the biopsy report (text phrases are listed in Supplementary Table 10) and notation of an established hematological cancer or hematological diagnosis upon manual review of their first available biopsy report. After exclusions, there were 117 participants in the final cohort without a known diagnosis of malignancy who underwent a first biopsy for an indication that included a WBC count.

(3) Taxane cohort: Participants were from a curated longitudinal cohort of 3492 BioVU participants undergoing treatment for primary cancer with taxane chemotherapies (paclitaxel or docetaxel)[20]. The cohort was originally constructed to examine differences in rates of incident drug-induced neutropenia between participants labeled as Black ($n = 365$) and White ($n = 3019$) race in their medical records. Analyses were restricted to 1724 European ancestry participants with existing genotyping and a baseline white blood cell ($n = 1721$) or neutrophil count ($n = 1724$) > 1000 cells/μL.

(4) Azathioprine cohort: Participants were from a curated multi-ancestry longitudinal cohort of 1466 BioVU participants with auto-immune or other inflammatory diseases who were newly started on the drug azathioprine[10]. The cohort was constructed to identify genetic factors associated with incident discontinuation of azathioprine among participants of black ($n = 165$) and white races ($n = 1301$). The analyses were restricted to the subset of 1180 European ancestry participants ages 20–80 years.

**eMERGE cohort.** The eMERGE consortium is a collection of institutions with DNA biobanks that are linked to electronic health records. These analyses examined adults of White European ancestry born before 1990 from the eMERGE network (phases 1--3)[35]. The participating eMERGE sites were Columbia University, Geisinger, Marshfield Clinic, Northwestern University, Mayo Clinic, Harvard University, Mt. Sinai Health System, and Kaiser Permanente/University of Washington, Seattle, and were approved by each eMERGE institution's IRB[35]. Participants with an ICD code for a hematological malignancy, chemotherapy, or radiation therapy (ICD codes listed in Supplementary Data 1) were excluded leaving 18,218 ($n = 10,162$ [56%] female) adult participants of European ancestry for these analyses.

**Michigan Genomics Initiative (MGI) cohort.** Study participants are derived from a biobank of patients recruited through the Michigan Medicine health system[36]. As of 2/2023, approximately 91,000 patients have consented to the linkage of a DNA sample with their Michigan Medicine electronic health record for research purposes. This analysis was performed on the Freeze 5 dataset of MGI ($n = 70,266$) and restricted to 59,910 samples of European ancestry based on genetic principal component analysis. Participants were selected for inclusion if they had an ICD code for inflammatory disease (inflammatory bowel disease or connective tissue diseases) and had azathioprine on their medication list after receiving the ICD code. Participants were excluded if they had an ICD code for solid organ transplant, a hematological malignancy. or a stem cell transplant. There were 417 participants after these exclusions. Participants were further excluded if they did not have a WBC count measure taken after azathioprine initiation. The final set comprised 354 participants.

**Genetic data**
BioVU participants were genotyped on the Illumina Infinium MEGA$^{EX}$ platform. SNP genotyping was called by the Vanderbilt Technologies for Advanced Genomics Analysis and Research (VANTAGE) Design core[37,38]. Genetic ancestry among BioVU participants was defined by genetic principal components (PCs) analysis in conjunction with HapMap reference populations by fitting PCs to a combined dataset of HapMap and BioVU participants. Using BioVU participants with an EHR-assigned race of White, the median value and the interval 4 standard deviations around the median were determined for the first two PCs. BioVU participants whose PC values fell within this interval were included in this study. Principal component plots visualizing the HapMap and BioVU populations are presented in (Supplementary Fig. 6). Participants were excluded for outlying heterozygosity measures, a discordance between genetically determined and reported sex, or >4% missing genetic data. Prior to imputation, data were put through the HRC-1000G check tool (v4.2.5) and pre-phased using Eagle v2.4.1[39]. QC analyses were performed using PLINK v1.90b6.17 and v2.00a3LM[40]. Imputation was performed using the Michigan Imputation Server in conjunction and the HRC v1.1 reference panel[41]. Imputed data were filtered for a sample missingness rate <2%, an SNP missingness rate <4%, and SNP deviation from Hardy-Weinberg $p < 10^{-6}$. PCs of ancestry were calculated across the entire BioVU cohort using the SNPRelate package[42].

eMERGE participants were genotyped on multiple platforms and underwent a similar QC analysis and imputation protocol as BioVU participants, as previously described[43,44].

MGI samples are genotyped in waves based on the time of recruitment, with initial waves genotyped on a customized Illumina Infinium CoreExome genotyping array and subsequent waves on a customized Illumina Infinium Global Screening Array. Genotypes are then imputed to 307,883,040 variants using the Trans-Omics for Precision Medicine (TOPMed) haplotype reference panel. 50,463,429 variants passed standard post-imputation filters, which removed poorly imputed variants with $r^2 < 0.3$ and very rare variants with minor allele frequency (MAF) < 0.01%.

**Development of a benign WBC polygenic score (PGS$_{WBC}$)**
The polygenic score (PGS) was derived from WBC count summary statistics from the European Ancestry subset of a genome-wide association study (GWAS) of hematological traits among ~750,000 participants[15]. A LD-reduced set ($r^2 < 0.01$) of non-palindromic SNPs associated with WBC count ($p < 5 \times 10^{-6}$, minor allele frequency = 0.01, imputation $r^2 \geq 0.7$) was selected using a clumping algorithm[45]. The $p$-value threshold was selected because it had the highest linear partial correlation ($r = 0.29$, adjusted for age and sex) between the PGS and measured WBC counts among the three thresholds examined ($p < 5 \times 10^{-8}$, $5 \times 10^{-7}$, $5 \times 10^{-6}$) in a set of 11,694 participants from BioVU without underlying hematological disease (the WBC Cohort described above).

To reduce the likelihood that the PGS included SNPs that are associated with clinically important diseases that cause a low WBC count, we first excluded all SNPs located in the Major Histocompatibility Complex genomic region (6:25500000–33500000), which is associated with multiple autoimmune diseases. We next identified all diagnoses reported in the GWAS Catalog[46] that reported an SNP associated with a phenotype at $p < 5 \times 10^{-7}$ and where that SNP was also associated with a WBC count at $p < 5 \times 10^{-6}$. There were 1507 phenotypes with at least 1 SNP that met these criteria. A manual review of these phenotypes identified 21 hematological malignancies and systemic lupus erythematous, all of which are important in the differential diagnosis of a low WBC count. These diagnoses are listed in Supplementary Table 11[17]. We identified all SNPs in the GWAS Catalog associated with each of these identified phenotypes at $p < 5 \times 10^{-7}$. We then excluded all SNPs in the PGS that were in LD ($r^2 > 0.5$) with one of these SNPs. After all exclusions, there were 1739 WBC-associated SNPs in the PGS$_{WBC}$. A complete listing of SNPs and weights comprising the PGS$_{WBC}$ can be found in Supplementary Data 2.

A weighted PGS$_{WBC}$ was calculated for each participant by summing the product of the allele dosage and the SNP weighting from the WBCs GWAS for each SNP. The distribution of the PGS$_{WBC}$ for the BioVU population and the individual cohorts is presented in Supplementary Fig. 7.

**Measured white blood cell counts**
For the WBC cohort, WBC counts collected on the same day as a routine health visit were extracted. WBC counts >35,000 cells/μL were excluded, as these values likely represent an active disease process. Each WBC count measure had an associated reference range that was specific to the clinical assay, with a lower bound typically of ~3900 cells/μL. A participant was labeled as an outlier if they had a WBC count measure below the lower bound indicated by the assay.

**ICD code-based phenotypes**
Phecodes are collections of related ICD-9/ICD-10 diagnosis codes (Phecodes definitions can be found at https://phewas.mc.vanderbilt.edu/)[47,48]. For each Phecode, cases are participants with one or more instances of the relevant ICD codes appearing in their medical records. Controls are participants without those codes and whose age fell

within the range of ages observed among cases. The Phecodes codes examined were for a low WBC count and low neutrophil count (neutropenia), and codes related to hematological malignancies ($n = 15$) and autoimmune diseases ($n = 21$) that were prevalent ($n > 100$ cases) among the BioVU participants. The list of Phecodes can be found in Supplementary Table 1.

For the phenotype of toxicity related to antineoplastic and immunosuppressive medications, cases were participants having any of the following ICD-9 (963.1, 960.7, 284.11, E933.1, E930.7) or ICD-10 (T45.1X4D, T45.1X4S, T45.1X5D, T45.1X5S, T45.1X5A, T45.1X1S, T45.1X4A, T45.1X2A, T45.1X1A) codes. To evaluate the specificity of the association for a low WBC count, cases were dichotomized into those with and those without a clinical note containing a mention of a low WBC count (keywords are listed in Supplementary Table 10) entered on the same day as the toxicity code.

**Bone marrow biopsy phenotypes**

Among 922 BioVU participants with a bone marrow biopsy report in their clinical record and without a prior history of hematological cancer or chemotherapy, there were 117 participants biopsied for a low WBC count (the Biopsy Cohort). The first biopsy report was reviewed, and extracted data included hematological comorbidities (related to platelets and red blood cell counts), other comorbidities listed in the clinical history, and the WBC count measured at the time of the biopsy or, if not available, the WBC count measured closest to the time of the biopsy. The primary outcome was a determination by the pathologist as to whether a clinically significant marrow abnormality was present (coded as Normal" or Abnormal)[12]. Data were extracted by a physician (JDM) and hematologist (SCB) to a REDCap (v13.*) database.

**Pharmacogenomic phenotypes**

For the taxane longitudinal study, the primary outcome was the development of a WBC count < 3000 cells/μL during the first cycle of treatment. Participants were censored at the earlier of a primary outcome event, the start of their second cycle of chemotherapy, or 1 month after the initiation of treatment. Secondary analyses examined an incident outcome of absolute neutrophil count < 1500 cells/μL.

For the MGI azathioprine study, the primary outcome was an incident WBC count < 3000 cells/μL. In secondary analyses, thresholds of <3500 and <4000 cells/μL were also examined to demonstrate that associations were consistent when using higher thresholds where larger numbers of participants met the threshold criteria. Participants were censored from the study at the earlier of a primary outcome event, or their last clinical encounter up to 24 months. The PGS<sub>WBC</sub> comprised a subset of 1704 SNPs that passed quality control.

For the BioVU azathioprine discontinuation study, the primary outcome was azathioprine discontinuation due to leukopenia or neutropenia, based on a provider's assessment, within 24 months of drug initiation. All participants were censored from the study at the earliest of a primary outcome event, time of drug discontinuation, or their last clinical encounter up to 24 months. In this separately genotyped population, the PGS<sub>WBC</sub> comprised a subset of 1,680 SNPs that passed quality control.

**Analysis**

A priori sex-stratified analyses were not specified, and post-hoc sex-stratified were not performed due to the small number of outcomes among the phenotypes examined.

The PGS<sub>WBC</sub> was mean standardized, and association statistics reflect changes per standard deviation (s.d.) increase in the PGS<sub>WBC</sub>. The PGS<sub>WBC</sub> was standardized separately within the BioVU, MGI, and BioVU azathioprine cohorts. A lower value reflects a polygenic predisposition toward lower WBC counts, and an inverse association between the PGS<sub>WBC</sub> and an outcome indicates that a predisposition to lower counts increases the likelihood of the outcome.

In the full BioVU cohort, multivariable logistic regression was used to determine whether the PGS<sub>WBC</sub> was associated with having a PheWAS code-based diagnosis of 15 bone marrow malignancies and 21 rheumatological diseases prevalent in this cohort. Models were adjusted for age, sex, and 5 PCs of genetic ancestry. Nominal association $p$-values are reported.

To determine whether the PGS may comprise SNPs associated with hematological malignancies, we also examined associations between 5 PGS constructed from subsets of SNPs based on quintiles of the distribution of effect sizes. Associations with each quintile-PGS and the 15 hematological malignancies were examined (as described above). Associations were also tested with 1000 permutations of each quintile-PGS to determine the empiric probability that a quintile-PGS had more nominal (association $p < 0.05$) phenotype associations than would be expected by chance. An empiric probability <0.05 was considered significant. We also examined the associations between the SNPs comprising each of the quintile-PGS and the 15 hematological malignancies and used binomial probabilities to identify phenotypes that had more nominal (association $p < 0.05$) associations than would be expected. The number of phenotypes with a binomial probability <0.05 was tallied for each quintile-PGS.

**WBC cohort.** For participants with multiple WBC count measures, the median value was used. The observed distribution of median WBC counts across the range of PGS<sub>WBC</sub> values was visualized by computing the 5th, 50th, and 95th percentiles of WBC count within sequential windows (±0.2 s.d.) across the range of the PGS<sub>WBC</sub>. Multivariable logistic regression, adjusting for age, age-squared, sex, and 5 PCs was used to test the association between the PGS<sub>WBC</sub> and (1) having a WBC count that fell outside the reference range and (2) having a Phecode for a low WBC count (Phecode 288.1). A Hosmer–Lemeshow test of goodness of fit based on 8 bins was used to assess for model goodness of fit. The association with the Phecode was also tested in the eMERGE cohort. In sensitivity analyses, SNPs in the PGS<sub>WBC</sub> that had an association with any of the 15 hematological malignancies (see above) were excluded (using nominal association thresholds ranging from 0.05 to 0.001) from the PGS, and associations with the two outcomes were re-evaluated.

Multivariable logistic regression, adjusting for age, sex, and 5 PCs, was used to determine whether the PGS<sub>WBC</sub> was associated with having a bone marrow biopsy for a clinical indication of low WBC count, among the 922 participants with a bone marrow biopsy.

Bone marrow biopsy cohort: Multivariable logistic regression was used to test the association between the PGS<sub>WBC</sub> and having a normal bone marrow biopsy finding, adjusting for age, sex, 5 PCs, and hematological comorbidities (low platelets, low red cells, other hematological abnormalities). In sensitivity analyses, SNPs in the PGS<sub>WBC</sub> that had an association with any of the 15 hematological malignancies (see above) were excluded (using nominal association thresholds ranging from 0.05 to 0.001) from the PGS, and associations with having a normal biopsy finding were re-evaluated.

**Bone marrow biopsy cohort.** In secondary analyses, we examined the number of biopsied individuals whose WBC count at the time of biopsy would lie within the expected range of values of a genotype-informed reference range. Specifically, individuals whose WBC count fell at or above the 2.5th percentile of the distribution of WBC counts observed among participants in the WBC cohort whose PGS<sub>WBC</sub> value was within 0.2 s.d. of that of the biopsied individual were classified as falling within their genetic reference range. For these analyses, the distribution of WBC counts in the WBC cohort was based on an individual's lowest observed value. Analyses were restricted to individuals with a WBC count at biopsy ranging from 0 to 4000 cells/μL, as individuals above these thresholds would be considered within

the normal ranges using standard thresholds. Analyses were stratified by whether the biopsy findings were normal or abnormal.

In the full BioVU cohort, multivariable logistic regression was used to determine whether the PGS_WBC was associated an ICD-based diagnosis of drug toxicity, with or without mention of a low WBC count. Models were adjusted for age, sex, and 5 PCs.

**Taxane and azathioprine cohorts.** Multivariable linear regression was used to determine the association between PGS_WBC and baseline log-transformed WBC count and change in WBC count (count at baseline - count at censoring time). A Cox proportional hazards regression model was used to estimate the hazard ratio associated with either incident leukopenia/neutropenia (taxanes) or azathioprine discontinuation for low counts. All analyses were adjusted for sex, age at drug initiation, and either 5 (taxane and azathioprine) or 10 (azathioprine discontinuation) PCs. The fit of the Cox models was assessed using analyses of Schoenfeld (proportional hazards assumption), Martingale (non-linearity), and deviance residuals (outliers). Data were visualized by Kaplan–Meier analysis.

All statistical tests were two-sided and a $p < 0.05$ was considered significant unless otherwise noted. Statistical analyses were conducted using R 4.2.0.

### Reporting summary

Further information on research design is available in the Nature Portfolio Reporting Summary linked to this article.

## Data availability

The BioVU subject-level data are available under restricted access based on the requirements of the participant consent process. Access to BioVU clinical and genetic data is controlled by the BioVU data repository (https://victr.vumc.org/biovu-description/#). Upon publication, data sets of individual-level phenotype data and corresponding data dictionaries to replicate the primary findings for the bone marrow biopsy outcomes, taxane, and azathioprine studies presented here for research purposes will be made available upon request from the repository (biovu@vumc.org). BioVU vetting for the use of individual-level data includes institutional IRB approval, data use agreements, and administrative and scientific reviews. eMERGE data are available through dbGaP (phs001584.v2.p2) and a list of deposited data and links can be found at https://emerge-network.org/dbgap/. Additional eMERGE phenotype data can be requested at https://emerge-network.org/contact/. The data generated in this study related to bone marrow biopsy outcomes, taxane treatment, and azathioprine discontinuation are provided in the Supplementary Information/Source Data file. Source data are provided with this paper.

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

## Acknowledgements

The authors acknowledge the MGI participants, Precision Health at the UM, the UM Medical School Central Biorepository, and the UM Advanced Genomics Core for providing data and specimen storage, management, processing, and distribution services and the Center for Statistical Genetics in the Department of Biostatistics at the School of Public Health for genotype data curation, imputation, and management in support of the research reported in this publication. This work was supported by the NIH R01GM130791 (J.D.M.), R01GM126535 (C.P.C.), R35GM131770 (C.M.S.), U01HG011181 (D.M.R.) and the Ingram Cancer Research Professorship fund (X.S.). Vanderbilt University Medical Center's BioVU is supported by institutional funding, private agencies, and federal grants. These include the NIH-funded Shared Instrumentation Grant S10RR025141; and CTSA grants UL1TR002243, UL1TR000445, and UL1RR024975. Genomic data are also supported by investigator-led projects that include U01HG004798, R01NS032830, RC2GM092618, P50GM115305, U01HG006378, U19HL065962, R01HD074711; and additional funding sources listed at https://victr.vumc.org/biovu-funding/. REDCap is supported by UL1TR000445 from NCATS/NIH. eMERGE is funded by U01HG006828 (Cincinnati Children's Hospital Medical Center/Boston Children's Hospital); U01HG006830 (Children's Hospital of Philadelphia); U01HG006389 (Essentia Institute of Rural Health, Marshfield Clinic Research Foundation, and Pennsylvania State University); U01HG006382 (Geisinger Clinic); U01HG006375 (Group Health Cooperative/University of Washington); U01HG006379 (Mayo Clinic); U01HG006380 (Icahn School of Medicine at Mount Sinai); U01HG006388 (Northwestern University); U01HG006378 (Vanderbilt University Medical Center); U01HG006385 (Vanderbilt University Medical Center serving as the Coordinating Center), U01HG004438 (CIDR) and U01HG004424 (the Broad Institute) serving as Genotyping Centers.

## Author contributions

J.D.M. had full access to all data in the trial and takes responsibility for the integrity of the data and the accuracy of the data analysis. J.D.M., A.L.D., J.Z., N.S.Z., C.M.S., S.L.V., S.C.B., X.S., C.M.C. and V.K.K. provided substantial contributions to the design of the study. L.D., N.S.Z., L.B., W.Q.W., M.S., G.P.J., E.A.R., A.K., A.S., I.J.K., T.L.W., J.G., H.H., N.J.C., D.M.R., S.C.B., X.S., S.G.F., B.V. and C.P.C. made substantial contributions to the acquisition of data. J.D.M. performed the primary analysis of the BioVU data. M.Z. performed analyses of the MGI data. J.D.M., C.M.S., J.P.S. and V.K.K. wrote the first draft of the manuscript. All co-authors critically reviewed the manuscript for important intellectual content, provided final approval of the version to be published, and agree to be accountable for all aspects of the work presented.

## Competing interests

S.C.B. has served on the scientific advisory board for Ipsen Pharmaceuticals and Fennec Pharmaceuticals. The remaining authors declare no competing interests.

## Additional information

[1]Department of Medicine, Vanderbilt University Medical Center, Nashville, TN, USA. [2]Department of Biomedical Informatics, Vanderbilt University Medical Center, Nashville, TN, USA. [3]Yale School of Medicine, New Haven, CT, USA. [4]Department of Genome Sciences, University of Washington Medical Center, Seattle, WA, USA. [5]Department of Medicine (Medical Genetics), University of Washington Medical Center, Seattle, WA, USA. [6]Division of Nephrology, Dept of Medicine, Vagelos College of Physicians & Surgeons, Columbia University, New York, NY, USA. [7]Department of Cardiovascular Medicine, Mayo Clinic, Rochester, MN, USA. [8]Department of Medicine, Northwestern University Feinberg School of Medicine, Chicago, IL, USA. [9]Department of Pediatrics, Children's Hospital of Philadelphia, Philadelphia, PA, USA. [10]Department of Pediatrics, Perelman School of Medicine, University of Pennsylvania, Philadelphia, PA, USA. [11]Department of Pharmacology, Vanderbilt University Medical Center, Nashville, TN, USA. [12]Department of Anesthesiology, University of Michigan Medical School, Ann Arbor, MI, USA. [13]Department of Biostatistics, Center for Statistical Genetics, University of Michigan, Ann Arbor, MI, USA. [14]Department of Pediatrics, Vanderbilt University Medical Center, Nashville, TN, USA. [15]Vanderbilt-Ingram Cancer Center, Vanderbilt University School of Medicine, Nashville, TN, USA. [16]Department of Medicine, University of Miami, Miami, FL, USA. ✉e-mail: jonathan.d.mosley@vumc.org

