## [Peer Review File · Nature Communications]

Clinical associations with a polygenic predisposition to benign lower white blood cell countsEditorial Note: This manuscript has been previously reviewed at another journal that is not operating a transparent peer review scheme. This document only contains reviewer comments and rebuttal letters for versions considered at *Nature Communications* .

REVIEWER COMMENTS

Reviewer #4 (Remarks to the Author):

The paper is written clearly, and the methods are adequately described. The conclusions are supported by the results. The authors have addressed the previous reviews well, in my opinion.

I only have some minor points that I would like the authors to address.

The SNPs and corresponding weights are provided in a supplementary table, allowing for reproducibility. Two minor points concerning this table:

- i) it would be helpful if the genome build was specified in the Position column of this table for added clarity.
- ii) It is "effect" with an e not an a. Please correct in columns F, G and H.

In the methods, when referring to the Taxane cohort, the authors note that it was constructed to examine racial differences in rates. Please ensure that the use of race is appropriate here. Refer to the NASEM Report on the use of such terms from earlier this year.

On a related note with regard to population descriptor language, I think that the language used in the "Genetic data" methods section on the selection of participants based on PCs should be written in a more precise manner. The authors used HapMap reference populations and excluded individuals who did not fall within 4 SDs of the medians of the first 2 PCs for "White European populations". It is unclear to me why the authors have chosen to use the term "White European" here. Would it not be more precise to refer directly to the HapMap reference for the medians, e.g. "CEU" etc.?

Reviewer #5 (Remarks to the Author):

I had a chance to review this revised manuscript by Mosley and colleagues as a new reviewer. I believe the authors have addressed most of the concerns raised in the initial review. I did find the manuscript and the main analyses a bit difficult to follow at times. It would be helpful to modify the flow to more clearly provide the rationale for all of the analyses presented. Moreover, given an extensive amount of work in the field of hematopoiesis, it would be good to discuss some of this in the context of these findings.

I have a few specific suggestions for improvement:

1. What is the distribution of PGS of subset cohorts/other cohorts compared to the initial BioVU cohort used to validate the PGS?

2. Page 7, second paragraph, last sentence: A similar association was seen in an independent set of 18,217 participants from the eMERGE network who did not have a history of a hematological malignancy, where 256 (1.4%) participants had an ICD code for a low WBC count (OR=0.74 [0.67 - 0.82], $p=2.0 \times 10^{-9}$).

If this is from previous work, there is no citation. If this is an analysis they performed, did they forget the in-text citation?

3. Page 9, first paragraph, first sentence: Bone marrow pathology was identified in 35 (30%) biopsies and was more frequent among participants who had other hematological comorbidities (e.g. anemia or a low platelet count) in addition to a low WBC count.

How many participants had low WBC count with and without other hematological comorbidities? What are the numbers in each of the groups?

4. Page 9, last paragraph: can you make a figure to visualize the results from the analysis you did in response to the reviewer?

5. Phrasing: Generally, I found the paper to be very confusing to read. I had to go back and re-read sentences multiple times. Sentences are too long and could often be split into two or more sentences. Here are some examples where things could be improved:

a. Page 6, last paragraph, second sentence:

Current: The PGSWBC was positively correlated (partial correlation=0.29, adjusted for age, sex and 5 PCs) with measured median WBC counts (Figure 2a), (Associations between the PGSWBC and an individual's lowest measured WBC count are presented in Supplementary Figure 2.)

Suggested: The PGSWBC was positively correlated with measured median WBC counts (partial correlation=0.29, adjusted for age, sex and 5 PCs) (Figure 2a). We also investigated the associations between the PGSWBC and an individual's lowest measured WBC count and found the same association (Supplementary Figure 2).

b. Page 7, first sentence:

Current: When WBC measurements are paired with the reference range specific to the clinical assay, there were 623 participants with at least one WBC count that fell below the lower reference range (i.e., a value that would be designated as a clinical outlier)

Suggestion: Please simplify this sentence.

c. Page 7, second sentence:

Current: The PGSWBC value was inversely associated with having a WBC count that fell below the reference range (odds-ratio [OR]=0.57 [95% CI: 0.52 – 0.62]) per s.d. increase in the PGSWBC, $p < 2 \times 10^{-16}$ (Figure 2b).

Suggestion: Please explain in lay terms as well – individuals with lower PGSWBC had an increased likelihood of having a WBC count below the reference range?

d. Page 7, paragraph 2, sentence 4:

Current: A lower PGSWBC was significantly associated with having an ICD code in BioVU (OR=0.62 [0.56 – 0.69], $p < 2 \times 10^{-16}$).

Suggestion: Specify having an ICD code for low WBC count. Earlier sentences in this paragraph could be improved too.

e. Page 8, last sentence:

Current: The PGSWBC was associated with having a biopsy for this indication (OR=0.56 [0.45 - 0.68], $p = 1.8 \times 10^{-8}$).

Suggestion: Please explain in lay terms as well – a lower PGSWBC is associated with having a biopsy for an WBC-count related indication?

f. Page 11, first paragraph, second sentence:

Current: The mean baseline (pre-treatment) WBC count was 8,200 (s.d. 3,900) cells/ μ L and the PGSWBC was significantly associated with baseline counts (log change in log[WBC count] per s.d. increase in the PGSWBC=0.062 [0.044 -0.080], $p = 3.9 \times 10^{-11}$).

Suggested: The mean baseline (pre-treatment) WBC count was 8,200 (s.d. 3,900) cells/ μ L. The PGSWBC was significantly associated with baseline (pre-treatment) WBC counts (logarithmic scale, PGSWBC=0.062 [0.044 -0.080], $p = 3.9 \times 10^{-11}$).

6. It is unclear from the sentence, "To ensure that SNPs associated with clinically significant diseases were not included in the PGSWBC, SNPs located in the major histocompatibility complex (MHC) genomic region, which is associated with multiple auto-immune diseases,¹⁶ or near loci associated with hematological malignancies or systemic lupus erythematosus¹⁷ were excluded (see methods for full details)," what hematologic malignancy associated loci were used. This is not clearly described in

the methods. Did the authors use very recent studies, such as dois: [10.1038/s41586-020-2786-7](https://doi.org/10.1038/s41586-020-2786-7), [10.1038/s41467-023-41315-5](https://doi.org/10.1038/s41467-023-41315-5), [10.1038/s41588-022-01121-z](https://doi.org/10.1038/s41588-022-01121-z)? If not, this analysis should be done.

Reviewer #4 (Remarks to the Author):

The paper is written clearly, and the methods are adequately described. The conclusions are supported by the results. The authors have addressed the previous reviews well, in my opinion. I only have some minor points that I would like the authors to address.

Response: We appreciate the reviewer's evaluation of our manuscript in the context of the prior revisions and suggestions for clarity and accuracy.

1. The SNPs and corresponding weights are provided in a supplementary table, allowing for reproducibility. Two minor points concerning this table:

- i) it would be helpful if the genome build was specified in the Position column of this table for added clarity.**
- ii) It is "effect" with an e not an a. Please correct in columns F, G and H.**

Response: We have corrected the incorrect spelling and noted the genome build (h19).

2. In the methods, when referring to the Taxane cohort, the authors note that it was constructed to examine racial differences in rates. Please ensure that the use of race is appropriate here. Refer to the NASEM Report on the use of such terms from earlier this year.

Response: We appreciate the reviewers point of clarity. This study was originally designed as an epidemiological study that evaluated drug discontinuation in the context of racial labels ("white" and "black") as assigned in the electronic health record. Ancestry was later determined when the population was genotyped. We have modified the text to clarify this point (page 20, paragraph 2).

"Taxane cohort: Participants were from a previously curated longitudinal cohort of 3,492 BioVU participants undergoing treatment for a primary cancer with taxane chemotherapies (paclitaxel or docetaxel).²¹ The cohort was originally constructed to examine differences in rates of incident drug-induced neutropenia between participants of black (n=365) and white (n=3019) races."

3. On a related note with regard to population descriptor language, I think that the language used in the "Genetic data" methods section on the selection of participants based on PCs should be written in a more precise manner. The authors used HapMap reference populations and excluded individuals who did not fall within 4 SDs of the medians of the first 2 PCs for "White European populations". It is unclear to me why the authors have chosen to use the term "White European" here. Would it not be more precise to refer directly to the HapMap reference for the medians, e.g. "CEU" etc.?

Response: We have modified the description in the Methods of the manuscript to better describe the approach used to select the BioVU population for these analyses. We have also added a new figure to visualize the BioVU population and the HapMap reference populations (please see new **Supplementary Figure 6**). (We agree that the term "White European" is not meaningful and have removed this.) The modified methods now read: (page 21, paragraph 3).

"Genetic ancestry among BioVU participants was defined by genetic principal components (PCs) analysis in conjunction with HapMap reference populations by

fitting PCs to a combined data set of HapMap and BioVU participants. Using BioVU participants with an EHR-assigned race of “white”, the median value and the interval 4 standard deviations around the median was determined for the first two PCs. BioVU participants whose PC values fell within this interval were included in this study. Principal component plots visualizing the HapMap and BioVU populations are presented in (**Supplementary Figure 6**).”

Reviewer #5 (Remarks to the Author):

I had a chance to review this revised manuscript by Mosley and colleagues as a new reviewer. I believe the authors have addressed most of the concerns raised in the initial review. I did find the manuscript and the main analyses a bit difficult to follow at times. It would be helpful to modify the flow to more clearly provide the rationale for all of the analyses presented. Moreover, given an extensive amount of work in the field of hematopoiesis, it would be good to discuss some of this in the context of these findings.

Response: We appreciate the reviewer's careful reading of our work and thoughtful comments regarding clarity and related to thinking about the construction of our genetic instruments.

I have a few specific suggestions for improvement:

1. What is the distribution of PGS of subset cohorts/other cohorts compared to the initial BioVU cohort used to validate the PGS?

Response: We now include a new **Supplementary Figure 7** that shows the distribution of the PGS in each of the BioVU cohorts examined, including the validation cohort. Note that for the bone marrow biopsy cohort, this cohort is restricted to participants selected for a biopsy due to clinical concern for a low WBC count. Hence, the distribution is shifted toward to lower ranges, as compared to the other cohorts, which are unselected with respect to WBC count. We reference this figure in the Methods section describing the construction of the PGS (page 23, paragraph 3):

"The distribution of the PGS_{WBC} for the BioVU population and the individual cohorts is presented in **Supplementary Figure 7.**"

2. Page 7, second paragraph, last sentence: A similar association was seen in an independent set of 18,217 participants from the eMERGE network who did not have a history of a hematological malignancy, where 256 (1.4%) participants had an ICD code for a low WBC count (OR=0.74 [0.67 - 0.82], p=2.0x10⁻⁹). If this is from previous work, there is no citation. If this is an analysis they performed, did they forget the in-text citation?

Response: We have clarified the wording of the results to indicate that this was an analysis performed in support of the current work (page 7, paragraph 2):

"We also tested for an association in an independent set of participants from the eMERGE network, a consortium of institutions with EHR-linked biobanks. Among eMERGE participants who did not have a history of a hematological malignancy (n=18,217), there were 256 (1.4%) participants who had an ICD code for a low WBC count. The PGS_{WBC} was associated with this outcome (OR=0.74 [0.67 - 0.82], p=2.0x10⁻⁹)."

3. Page 9, first paragraph, first sentence: Bone marrow pathology was identified in 35 (30%) biopsies and was more frequent among participants who had other hematological comorbidities (e.g. anemia or a low platelet count) in addition to a low WBC count. How many participants had low WBC count with and without other hematological comorbidities? What are the numbers in each of the groups?

Response: This sentence is confusing as written, and we have now clarified it. We have also updated **Supplementary Table 5**, which provides the characteristics of the bone marrow cohort, to show the counts referred to in the modified text. The updated text of the manuscript now reads (page 8, paragraph 3):

“Bone marrow pathology was identified in 35 (30%) biopsies. Biopsies which showed pathology were more likely to have a second hematological comorbidity (e.g. anemia or a low platelet count) noted in the indication, as compared to those without pathology (n=29 [83%] vs n=37 [45%]) (**Supplementary Table 5**).”

4. Page 9, last paragraph: can you make a figure to visualize the results from the analysis you did in response to the reviewer?

Response: We have now added new **Figures 3c** and **3d** which shows the underlying data used to make the calculations shown in **Table 1**. These scatterplots show the WBC count and PGS_{WBC} combinations that led the count to be classified as within the normal range.

5. Phrasing: Generally, I found the paper to be very confusing to read. I had to go back and re-read sentences multiple times. Sentences are too long and could often be split into two or more sentences. Here are some examples where things could be improved:

Response: Thank you for the suggestions to clarify the text. We have gone through these suggestions (below) and have reviewed the manuscript to simplify sentences.

a. Current: The PGSWBC was positively correlated (partial correlation=0.29, adjusted for age, sex and 5 PCs) with measured median WBC counts (Figure 2a), (Associations between the PGSWBC and an individual’s lowest measured WBC count are presented in Supplementary Figure 2.)

Suggested: The PGSWBC was positively correlated with measured median WBC counts (partial correlation=0.29, adjusted for age, sex and 5 PCs) (Figure 2a). We also investigated the associations between the PGSWBC and an individual’s lowest measured WBC count and found the same association (Supplementary Figure 2).

Response: We have incorporated the reviewer’s suggestion (page 6, paragraph 3).

b. Current: When WBC measurements are paired with the reference range specific to the clinical assay, there were 623 participants with at least one WBC count that fell below the lower reference range (i.e., a value that would be designated as a clinical outlier)

Suggestion: Please simplify this sentence.

c. Current: The PGSWBC value was inversely associated with having a WBC count that fell below the reference range (odds-ratio [OR]=0.57 [95% CI: 0.52 – 0.62]) per s.d. increase in the PGSWBC, p<2x10⁻¹⁶) (Figure 2b).

Suggestion: Please explain in lay terms as well – individuals with lower PGSWBC had an increased likelihood of having a WBC count below the reference range?

Response: We have simplified these sentences as follows (page 6, paragraph 3).

“When WBC count measurements are reported in clinical settings, the results are presented in conjunction with the reference range for the assay used to measure the

count. There were 623 participants who had at least one WBC count that fell below the lower reference range value (i.e., these participants had a value that would be designated as a clinical outlier). The PGS_{WBC} was inversely associated with the outcome of having a WBC count below the lower reference range value (odds-ratio [OR]=0.57 [95% CI: 0.52 – 0.62] per s.d. increase in the PGS_{WBC} , $p < 2 \times 10^{-16}$) (Figure 2b).”

d. Current: A lower PGSWBC was significantly associated with having an ICD code in BioVU (OR=0.62 [0.56 – 0.69], $p < 2 \times 10^{-16}$).

Suggestion: Specify having an ICD code for low WBC count. Earlier sentences in this paragraph could be improved too.

Response: We have modified the paragraph as follows (page 7, paragraph 2).

“ICD codes are assigned to patients to indicate the diagnoses that were addressed during a clinical encounter.¹⁸ If a patient has been assigned an ICD code for a low WBC count, it indicates that the clinical provider diagnosed the patient with a low WBC count. There were 379 (3.2%) participants assigned an ICD code for a low WBC count. The PGS_{WBC} was significantly associated with being assigned ICD code for a low WBC count among BioVU participants (OR=0.62 [0.56 – 0.69], $p < 2 \times 10^{-16}$). Thus, a polygenic predisposition to lower WBC counts increased the likelihood of receiving a clinical diagnosis of a low WBC count.”

e. Current: The PGSWBC was associated with having a biopsy for this indication (OR=0.56 [0.45 - 0.68], $p = 1.8 \times 10^{-8}$).

Suggestion: Please explain in lay terms as well – a lower PGSWBC is associated with having a biopsy for an WBC-count related indication?

Response: We have simplified the entire paragraph to read (page 8, paragraph 2).

“There were 922 BioVU participants without a prior history of a hematological malignancy who underwent a first bone marrow biopsy. We tested whether the PGS_{WBC} was associated with undergoing a biopsy due to a clinical concern for a low WBC count. There were 117 participants biopsied for this reason, and the PGS_{WBC} was associated with this outcome (OR=0.56 [0.45 - 0.68], $p = 1.8 \times 10^{-8}$). This result indicates that a polygenic predisposition to a lower WBC count increased the likelihood that a biopsy was performed due to concerns for a low count.”

f. Page 11, first paragraph, second sentence:

Current: The mean baseline (pre-treatment) WBC count was 8,200 (s.d. 3,900) cells/ μ L and the PGSWBC was significantly associated with baseline counts (log change in log[WBC count] per s.d. increase in the PGSWBC=0.062 [0.044 -0.080], $p = 3.9 \times 10^{-11}$).

Suggested: The mean baseline (pre-treatment) WBC count was 8,200 (s.d. 3,900) cells/ μ L. The PGSWBC was significantly associated with baseline (pre-treatment) WBC counts (logarithmic scale, PGSWBC=0.062 [0.044 -0.080], $p = 3.9 \times 10^{-11}$).

Response: We have incorporated the reviewer’s suggestion (page 10, paragraph 3).

6a. It is unclear from the sentence, "To ensure that SNPs associated with clinically significant diseases were not included in the PGSWBC, SNPs located in the major

histocompatibility complex (MHC) genomic region, which is associated with multiple auto-immune diseases,¹⁶ or near loci associated with hematological malignancies or systemic lupus erythematosus¹⁷ were excluded (see methods for full details)," what hematologic malignancy associated loci were used. This is not clearly described in the methods.

Response: We have rewritten the methods section to better explain the methods. The list of hematological malignancies in the GWAS catalog that were used to exclude SNPs from the PGS are listed in **Supplementary Table 11**. The modified Methods read (page 23, paragraph 2).

"To reduce the likelihood that the PGS included SNPs that are associated with clinically important diseases that cause a low WBC count, we first excluded all SNPs located in the Major Histocompatibility Complex genomic region (6:255000000-335000000), which is associated with multiple auto-immune diseases. We next identified all diagnoses reported in the GWAS Catalog⁴³ that reported a SNP associated with a phenotype at $p < 5 \times 10^{-7}$ and where that SNP was also associated with a WBC count at $p < 5 \times 10^{-6}$. There were 1,507 phenotypes with at least 1 SNP that met these criteria. A manual review of these phenotypes identified 21 hematological malignancies and systemic lupus erythematosus, all of which are important on the differential diagnosis for a low WBC count. These diagnoses are listed in **Supplementary Table 11**.¹⁷ We identified all SNPs in the GWAS Catalog associated with each of these identified phenotypes at $p < 5 \times 10^{-7}$. We then excluded all SNPs in the PGS that were in LD ($r^2 > 0.5$) with one of these SNPs. After all exclusions, there were 1,739 WBC-associated SNPs in the PGS_{WBC}. A complete listing of SNPs and weights comprising the PGS_{WBC} can be found in **Supplementary Materials**."

6b. Did the authors use very recent studies, such as dois: 10.1038/s41586-020-2786-7, 10.1038/s41467-023-41315-5, 10.1038/s41588-022-01121-z? If not, this analysis should be done.

Response: We did not exclude SNPs associated with intermediate hematological cancer phenotypes, such as clonal hematopoiesis. Though we did exclude SNPs that were associated with myeloproliferative neoplasms that were described in the recent studies (e.g. 10.1038/s41588-022-01121-z). Per the reviewer's suggestion, we identified loci associated with the clonal hematopoiesis phenotypes. SNPs nominally associated with CH phenotypes at $p < 10^{-7}$ represented 25 loci. Of these, there were 3 SNPs in the PGS_{WBC} (of 1,739 SNPs) that tagged a lead SNP in these loci ($r^2 > 0.5$). (Of note, these 3 loci were identified from a multi-trait GWAS performed on the outcomes of loss of chromosome Y and telomere length.) In our sensitivity analyses, where we excluded a broader set of SNPs, these three SNPs were excluded from the PGS (e.g. see **Supplementary Tables 4 and 6**). The reviewer's comments bring up a larger, relevant issue with respect to attempting to construct a biomarker PGS that minimally measures genetic variation associated with important diseases. We have added a paragraph to the Discussion to discuss this issue (page 16, paragraph 3).

"A challenge of constructing a "benign" PGS for a clinical biomarker is determining which SNPs that may predispose to disease should be excluded from the predictor. For the PGS_{WBC}, we excluded SNPs in genomic loci that were associated with diseases that are in the differential diagnosis of a low WBC count, such as hematological malignancies. However, we did not exclude SNPs

associated with intermediate phenotypes of hematological malignancies, such as clonal hematopoiesis, as these phenotypes often will not progress to disease and are also typically not associated with WBC count abnormalities in the absence of transformation to malignant disease.³⁰⁻³³ More generally, when constructing a “benign” PGS for possible clinical application, it will be essential to define the clinical settings where the PGS will be used and to exclude all SNP variation associated with important diagnoses relevant to the setting and the biomarker. Defining the appropriate association p-value thresholds for exclusion can present challenges. Excluding a highly inclusive set of SNPs weakly associated with a large number of diseases will remove numerous SNPs from the PGS and degrade its performance. In contrast, only excluding SNPs associated with diseases at genome-wide significance may be too restrictive and risks misclassifying diseased individuals as benign outliers. Importantly, for any set of exclusion criteria, validation of the performance and utility of the PGS in real-world settings will be critical to determine its utility, safety and limitations in practice.”

REVIEWERS' COMMENTS

Reviewer #5 (Remarks to the Author):

The authors have addressed all of my concerns.

I did note one typo: Page 6, line 131, "that expected by chance" should be "than expected by chance"

It will be great to see this work published soon.

REVIEWERS' COMMENTS

Response to Reviewer #5:

The authors have addressed all of my concerns. It will be great to see this work published soon.

I did note one typo: Page 6, line 131, “that expected by chance” should be “than expected by chance”

Response: Thank you, again, for your careful reading. We have made the correction noted.